# Domain adaptation using optimal transport for invariant learning using histopathology datasets

**Kianoush Falahkheirkhah**[1]                                        KF4@ILLINOIS.EDU
[1] *Beckman Institute for Advanced Science and Technology, University of Illinois at Urbana-Champaign*
**Alex Lu**[2]                                                      LUALEX@MICROSOFT.COM
**David Alvarez-Melis**[2]                                    ALVAREZ.MELIS@MICROSOFT.COM
[2] *Microsoft Research, New England*
**Grace Huynh**[3]                                         GRACE.HUYNH@MICROSOFT.COM
[2] *Microsoft Research, Redmond*

**Editors:** Under Review for MIDL 2023

## Abstract

Histopathology is critical for the diagnosis of many diseases, including cancer. These protocols typically require pathologists to manually evaluate slides under a microscope, which is time-consuming and subjective, leading to interest in machine learning to automate analysis. However, computational techniques are limited by batch effects, where technical factors like differences in preparation protocol or scanners can alter the appearance of slides, causing models trained on one institution to fail when generalizing to others. Here, we propose a domain adaptation method that improves the generalization of histopathological models to data from unseen institutions, without the need for labels or retraining in these new settings. Our approach introduces an optimal transport (OT) loss, that extends adversarial methods that penalize models if images from different institutions can be distinguished in their representation space. Unlike previous methods, which operate on single samples, our loss accounts for distributional differences between batches of images. We show that on the Camelyon17 dataset, while both methods can adapt to global differences in color distribution, only our OT loss can reliably classify a cancer phenotype unseen during training. Together, our results suggest that OT improves generalization on rare but critical phenotypes that may only make up a small fraction of the total tiles and variation in a slide.

**Keywords:** Domain adaptation, Batch effect, Optimal transport, Digital pathology, Deep learning.

## 1. Introduction

Histopathology is a crucial tool in the diagnosis of various diseases, including cancer. In this process, a tissue sample is taken from a patient, thinly sliced, fixed to a microscope slide, and stained to visualize cells and tissue. By examining samples under a microscope, pathologists diagnose disease by identifying changes in cells and tissue (such as abnormal mitosis in cancer)(Gurcan et al., 2009). However, manual qualitative evaluation is time-consuming and subjective, leading to the development of various machine learning methods, including image segmentation and inference of molecular features and therapy response (Echle et al., 2021; Van der Laak et al., 2021).

Although initial models have shown promise, few algorithms have demonstrated robust performance in widespread clinical implementation. One reason for this is that pathology

machine learning models are challenged by batch effects (Howard et al., 2021). Different histopathology images may exhibit subtle differences stemming from variation in slide preparation, staining protocol, and data processing. While these effects are often systematic between institutions (e.g. institutions may use different scanners technologies to digitize their images), even slides prepared at the same institution can still be impacted by slight variations in preparation protocol. While human pathologists can largely ignore these differences, machine learning models are severely impacted, so accuracy degrades as models are deployed in new hospitals (Shaban et al., 2019; Lafarge et al., 2019).

Making histopathology models robust to batch effects is an active area of research, with several general strategies proposed. In this manuscript, we focus on domain invariant representation learning. Compared to other strategies that may require targeted domain knowledge to design augmentations (Faryna et al., 2021; Pérez-Bueno et al., 2022; Ataky et al., 2020) or access to test images for alignment or retraining (Lafarge et al., 2019; Abbet et al., 2021; Saenko et al., 2010), domain invariant representation learning aims to learn a model that is intrinsically robust to batch effects, by ensuring only biological variation (and not batch effects) is present in the representation learned by a model. The simplest way to achieve this premise is through use of an adversary during training (Ganin et al., 2016), by penalizing a model if hospital institution can be classified from the model's representation of images (Lafarge et al., 2017, 2019), under the premise that that data across institutions should not be substantially different in biological content, and differences stem mostly from technical batch effects.

Most previous research in histopathology that utilizes this strategy relies on a loss function that penalizes the ability to classify institutions from individual images. While this approach is technically simple, these losses are prone to failure because the adversary is prone to focus on the more frequent differences between datasets. While these distributional differences can be systematic (e.g. the intensity of stain can be systematically shifted), these systematic differences interact with more local subpopulation shifts to cause batch effects in the training dataset to be under-observed in certain phenotypes or tissues. For example, the training dataset may not have sufficient examples across the full distribution of biological variability, or differences in fixation may cause changes in some tissues and not others (Chatterjee, 2014).

Inspired by theoretical advances suggesting that these failures can be mitigated by considering batches of images —rather than individual instances– we propose a new optimal transport (OT) based loss. Our OT-based loss allows us to compare if the distributions of two sets of images are different, which we use to quantify and penalize representational differences in images between institutions. We demonstrate that our OT loss method improves overall performance on the Camelyon17 dataset over standard image-based adversarial models. At the tile level, we observe improved performance for those images containing features that are within the typical distribution of biological variation, but poorly represented in the training set.

## 2. Background

### 2.1. Optimal Transport

Optimal Transport is a mathematical framework to compare, align, and transform probability distributions. It was originally formulated by Gaspard Monge (Monge, 1781) who was interested in finding optimal matchings between source and target locations across which materials or goods had to be transported. Recently, a more general formulation has gained popularity in various fields including computer vision (Arjovsky et al., 2017), natural language processing (Alvarez-Melis and Jaakkola, 2018), and economics (Galichon, 2018).

In this work, we consider a classification setting where we are given labeled data $\{(x_i^s, y_i^s)\}$ from a source domain $(S)$, but only unlabeled data $\{x_j^t\}$ from the target domain $(T)$ is available. Here we assume the features of both datasets have the same dimensionality, e.g., $x_i^s, x_j^t \in \mathbb{R}^d$. The goal is then to leverage the data from the source domain to train a model that will ultimately be used in the target domain. In the machine learning literature, this is known as the *unsupervised* domain adaptation problem. Optimal transport can be used to define a distance between the source and target samples by treating them as discrete distributions supported on finitely many points. Formally, we define the distributions $\mu_s = \sum_{i=1}^{n_s} p_i^s \delta_{x_i^s}$ and $\mu_t = \sum_{i=1}^{n_t} p_i^t \delta_{x_i^t}$ where $\delta_{x_i}$ is the Dirac at $x_i$, and $p^s$ and $p^t$ are probability vectors associated with the samples. Given a cost function $c(\cdot, \cdot) : \mathbb{R}^d \times \mathbb{R}^d \to \mathbb{R}^+$, the Kantorovich (Kantorovitch, 1958) formulation of the discrete OT problem consists of finding a coupling matrix $\Gamma \in \mathbb{R}^{n_s \times n_t}$ minimizing:

$$\mathrm{OT}(\mu_s, \mu_t) = \min_{\Gamma \in \Pi(\mu_s, \mu_t)} \langle \Gamma, C \rangle, \tag{1}$$

where $\langle ., . \rangle$ is the inner product, $C_{ij} = c(x_i^s, x_j^t)$ is the cost associated with moving probability mass from $x_i^s$ to $x_j^t$ (typically taken to be the euclidean distance between these two points), and $\Pi$ is the probabilistic coupling between the distribution of $S$ and $T$ which is a set of doubly stochastic matrices defined as below:

$$\Pi(\mu_s, \mu_t) = \{\Gamma \in \mathbb{R}_+^{n_s \times n_t} \mid \Gamma 1 = \mu_s, \ \Gamma^T 1 = \mu_t\} \tag{2}$$

As Equation 1 is a linear programming problem, the runtime complexity of solving it scales cubically with the number of samples (Peyré et al., 2019), which is typically prohibitive in applications. Moreover, in the presence of outliers, this formulation may lead to incorrect transportation of points and demonstrate some irregularities. To overcome the aforementioned challenges, Cuturi (2013) introduced a regularized version of Equation 1 by adding an entropy term as below:

$$\mathrm{OT}(\mu_s, \mu_t) = \min_{\Gamma \in \Pi(\mu_s, \mu_t)} \langle \Gamma, C \rangle + \epsilon H(\Gamma) \tag{3}$$

Where $H(\Gamma) = -\sum_{i,j} \Gamma(i,j) \log \Gamma(i,j)$ computes the entropy of $\Gamma$. Equation 3 can be solved using the Sinkhorn-Knopp algorithm (Cuturi, 2013).

### 2.2. Optimal transport to regularize model training

We consider a simple tumor/normal classification model $(C)$ for histopathology data, and seek a way to generalize our model without knowledge of the target data distribution.

We first use a model ($\phi$) to map the raw histology images to a feature space, then a fully-connected neural network ($C$) is used to map the feature space to the classifications. We use our unlabeled validation data to define an optimal transport distance between the training and validation data at the feature space level. This is minimized simultaneously with the training of the classification model, to limit overfitting and promote invariant learning. At test time, we deploy only the classification model, without any additional domain adaptation steps.

Our overall loss function is:

$$L_{total} = L_{\text{CE}} + \alpha * L_{\text{OT}}, \tag{4}$$

where $\alpha$ is a hyperparameter controlling the strength of cross-domain regularization (discussed in detail in the results section) and $L_{CE}$ is a cross-entropy loss, i.e.:

$$L_{\text{CE}} = -\sum(\log(C(\phi(X_S; \theta_\phi); \theta_C)). \tag{5}$$

Here $X_S$ is a batch of samples from the source domain, and $\theta_\phi$ and $\theta_C$ are the weights of the featurizer $\phi$ and classifier $C$, respectively. Finally, $L_{OT}$ is the domain generalization defined as:

$$L_{\text{OT}} = \sum(\text{OT}(\phi(X_S, \theta_S), \phi(X_T, \theta_S))), \tag{6}$$

where $X_T$ is a batch of samples from the target domain (validation).

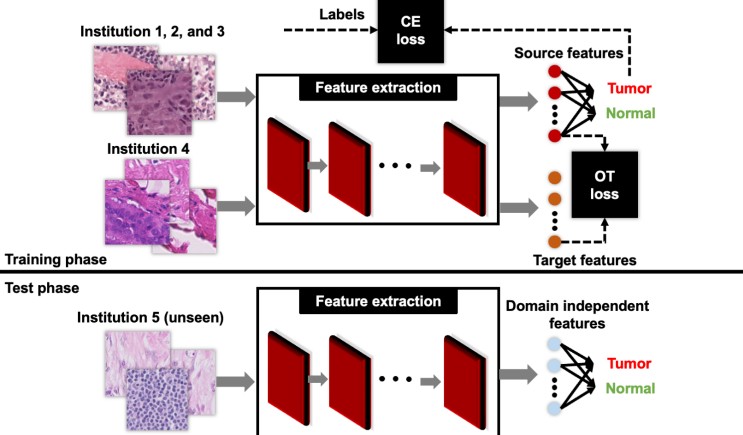

Figure 1: **Method overview**. Our proposed OT loss is integrated during model training. The total objective function during the training phase is a combination of a standard classification cross-entropy (CE) loss and OT loss, weighted by the tuning parameter $\alpha$. Training on two sets of institutions, the labeled set of institutions 1-3, and the unlabeled institution 4, leads to finding more robust, domain-invariant features. At test time, the model is simply evaluated using the domain-independent features.

## 2.3. WILDS dataset

In this study, we used the Camelyon17-WILDS benchmark dataset, designed to support method development for addressing distribution shift (Koh et al., 2021). Camelyon17-WILDS is a tiled and sampled version of the Camelyon-17 dataset, which contains Hematoxylin and Eosin (H&E) stained images of breast lymph node resections from 5 institutions (Litjens et al., 2018). A representative example is shown in Appendix A. The Camelyon17-WILDS dataset includes sparsely sampled 96x96 patches from a whole-slide image of a lymph node; each patch has been annotated by experts as tumor or non-tumor. Sample patches are 96x96 pixels, and the entire dataset has previously been partitioned into a training, validation and test set; there are a total of 302,436 patches from 30 WSI and 3 institutions in the training set, 34,904 patches from 10 WSI and 1 institution in the validation set, and 85,054 patches from 10 WSI and 1 institution in the test set.

## 2.4. Computational Implementation

The full code for our implementation can be found here: https://github.com/kiakh93/OT-regularized-UDA. For all experiments, we use ResNet-50 architecture initialized randomly. In addition, we substitute Batch Normalization layers with Instance normalization layers. We used a learning rate of $10^{-3}$, $L_2$ regularization with $10^{-3}$, and SGD optimizer with momentum of 0.9. All models were trained for 5 epochs with early stopping using 96 x 96 patches with batch size of 128. We report the results that are aggregated over 4 experiments with random initialization.

## 3. Methodology and Results

Our OT loss method can be used for unsupervised domain adaptation between two seen sets of institutions, one of which we have labels for, and the other for which we do not have labels. We explore this by using the labeled training dataset, institutions 1-3, and the unlabeled validation dataset, institution 4, shown in Figure 1. However, our goal is to develop a model that can generalize to unseen institutions, *without model retraining*. Our hypothesis is that learning to minimize between two sets of institutions will lead to invariant features that then generalize to future unseen domains without retraining. To do this, we minimize the OT loss during training on the two sets of seen institutions (training institutions 1,2,3 and validation institution 4), then simply evaluate the model, without retraining, on the test dataset (institution 5 in Figure 1).

We demonstrate our model on the Camelyon17-WILDS dataset, which seeks to classify tissue crops as normal or tumor under domain shift. Specifically, we trained a ResNet50 model (He et al., 2016) to classify images in the training dataset using a cross-entropy loss. During training, we also encoded validation images and calculated the OT distance between features at the final layer of the ResNet50 encoder immediately before the classification layer between validation batches and training batches. The overall loss of the classification model is a combination of the cross entropy loss and the OT loss, regularized by the tuning parameter $\alpha$. To tune $\alpha$, we selected the parameter achieving the highest accuracy for the validation dataset, $\alpha = 0.1$ (Table 1). Additional experiments on how parameterization impacts model training can be found in Appendix B.

Table 1: **Model performance using various $\alpha$, representing the OT contribution to the overall loss function**. We report standard deviation in the parenthesis

| $\alpha$ | 0.00001 | 0.0001 | 0.001 | 0.01 | 0.1 | 1 |
|---|---|---|---|---|---|---|
| **Accuracy of validation** | 0.882 (0.011) | 0.882 (0.002) | 0.871 (0.004) | 0.882 (0.007) | **0.891 (0.005)** | 0.712 (0.165) |
| **Accuracy of test** | 0.799 (0.029) | **0.857 (0.011)** | 0.811 (0.019) | 0.826 (0.012) | 0.850 (0.019) | 0.733 (0.150) |

Table 2: **Model performance compared to DANN method**. We report standard deviation in the parenthesis

| | OT-regularized | DANN |
|---|---|---|
| **Accuracy of validation** | 0.891 (0.005) | 0.873 (0.014) |
| **Accuracy of test** | 0.850 (0.019) | 0.796 (0.052) |

As a baseline, we compared our method against the domain adaptive neural network (DANN) method. Although DANN is one of the earliest deep learning-based approaches for domain adaptation, it continues to be utilized and remains a popular method for domain adaptation in many recent studies (Chen et al., 2022; Aminian et al., 2022). Our work is analogous to DANN in that it also achieves robustness by aligning the representation space of the model during training, penalizing the model if different domains can be distinguished, and can be used as a drop-in replacement. In DANN, individual images are penalized, while the OT loss penalizes batches of images. Our goal in comparing to DANN is to show that a drop-in replacement of the adversarial objective in DANN with an OT loss can lead to a more effective and nuanced correction. To maximize comparability, we trained a DANN model using the same architecture and data splits as our OT method. We observed that our OT method outperforms DANN on both the validation and test dataset, and with a greater margin on the test dataset (Table 2). ROC and AUC can be found in Appendix C. Finally, as a more trivial baseline, we compared if using the OT distance to align validation and test features with training features after training (i.e. with a frozen ResNet50) would be effective (Appendix D). This strategy leads to much inferior performance than either our OT method or DANN, suggesting that incorporating OT during training is critical to model generalization, rather than the strength of OT in correction alone.

To understand why our OT method outperforms DANN, we performed a qualitative evaluation of the training, validation, and test datasets using features extracted by a frozen ImageNet-trained encoder with tSNE (Figure 2). In addition, we provide t-SNE projection of OT-regularized and DANN in Appendix E. Unlike features learned by either our OT or the DANN method, ImageNet features are not trained on or exposed to any images in Camelyon17, allowing for an unbiased exploration of the perceptual similarity of tiles in these datasets. We observed that while tumor and normal tiles form distinct clusters in

the training dataset (clusters 1 and 4 in Figure 2), tumor and normal tiles now form a continuous single cluster in the test dataset (clusters 2,3, and 5 in Figure 2).

In addition, we note that many of the test tiles contain image features that are poorly represented in the training/validation set (gray/test points shifted rightward in Figure 2A to a region with few brown/pink/train/val points). Correspondingly, we visualize the qualitative differences in the pathology tiles across the training, validation, and test datasets in Figure 2D, and note that there are both differences in color/staining and differences in biological feature representation in the test clusters compared to the training clusters. These features, biological and non-biological, are all within the expected distribution representing typical variability in both tumor and normal images. We observe that particularly in the feature space of cluster 3, DANN has a much higher failure rate while the OT method succeeds, suggesting that our OT method may be better able to capture the full distribution of image variability on the feature representation level, even when there is poor representation in that feature space during model training. To demonstrate the robustness of our approach, we have switched the validation and test sets, using institution 5 as the validation set and institution 4 as the test set (unseen). Our results indicate (Appendix F ) that the OT-regularized model still performs well when using either institution as the unseen institution.

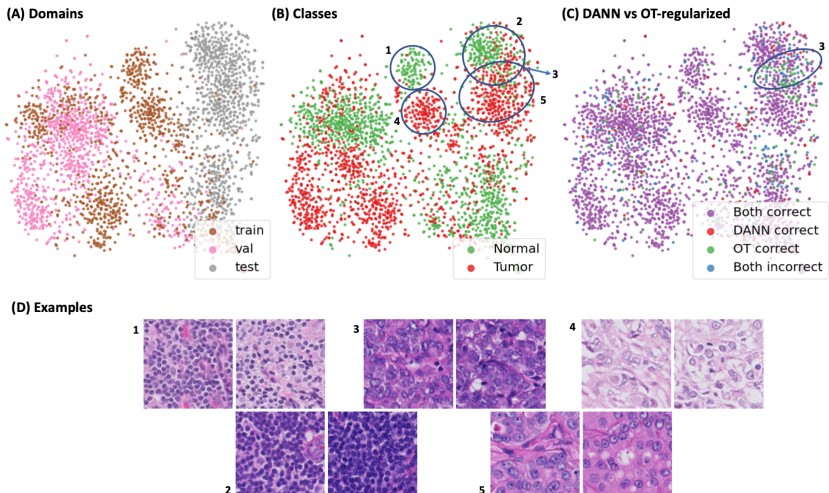

Figure 2: **The t-SNE projection of feature embeddings from a ResNet-50 model trained on ImageNet**. (A) represents the distribution of each domain. (B) highlights the distribution of normal/tumor patches. (C) compares the performance of DANN and OT. (D) demonstrate examples from the distributions that have been highlighted in (B).

Finally, while our previous results show that our proposed OT loss leads to significant improvements in generalization in adversarial methods, correcting batch effects in histopathology is a well-studied topic, and numerous other approaches have been proposed.

Table 3: **Model performance compared to benchmark algorithms.**. We report standard deviation in the parenthesis

| Methods | ERM | CAROL | IRM | Group DRO | MBDG | SGD (Freeze-Embed) | OT-regularized |
|---|---|---|---|---|---|---|---|
| **Accuracy of** | 0.849 | 0.862 | 0.862 | 0.855 | 0.881 | 0.952 | 0.891 |
| **validation** | (0.031) | (0.014) | (0.014) | (0.022) | (0.018) | (0.003) | (0.005) |
| **Accuracy of** | 0.703 | 0.595 | 0.642 | 0.684 | 0.933 | 0.965 | 0.850 |
| **test** | (0.064) | (0.077) | (0.081) | (0.073) | (0.010) | (0.004) | (0.019) |

We next took a more systematic comparison of the performance of our method against other algorithms previously reported on the WILDS-Camelyon17 dataset: empirical risk minimization (ERM), which minimizes the average training loss and does not include any domain adaptation, CORAL (Gulrajani and Lopez-Paz, 2021), originally designed for unsupervised domain adaptation; IRM (Arjovsky et al., 2019), for domain generalization; and Group DRO (Sagawa et al., 2019), for subpopulation shifts. These baselines indicate that domain adaptation is not trivial on histopathology images, with all methods worsening performance on the test dataset relative to the ERM baseline. Compared to these methods, our method performs significantly better than ERM.

More recent approaches in correcting batch effects have investigated the use of synthetic images (MBDG (Robey et al., 2021)), improved architectures like vision transformers (SGD (Freeze Embed) (Kumar et al., 2022))), and adaptive data augmentation (Appendix G). While these tailored approaches outperform our more general OT method in this setting, we highlight opportunities for synergy between approaches in the next section.

## 4. Discussion

In this paper, we show that using optimal transport to quantify and penalize domain differences during training leads to more invariant learning for histopathology image tiles, improving the likelihood that models can be directly deployed on new institutions without retraining. Unlike standard adversarial approaches, our method correctly classifies some test image tiles in regions where there are insufficient training examples. We hypothesize that this is because our OT loss is more robust to subpopulation shifts between the training and test sets in the image feature space, which may arise when the training set does not have sufficient examples across the full distribution of image feature variability, whether arising from biological or tissue processing causes (see Appendix G).

While our OT loss does not out-perform the top proposed batch effect correction methods on Camelyon17, the top current methods take different approaches than our method, indicating opportunities for synergy. For example, OT loss could be used in combination with stain augmentation methods that have proven effective in resolving institutional differences in color and intensity of histopathology samples, but do not address subpopulation shifts in the image feature space. Other studies, which have proposed more advanced architectures and synthetic data, could also synergize with our method and lead to better overall performance(Robey et al., 2021; Kumar et al., 2022). In conclusion, future work may explore how this method performs on larger fields-of-view, more challenging tasks, and more unseen domains.

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

## Appendix A. Example of patches in WILDS dataset

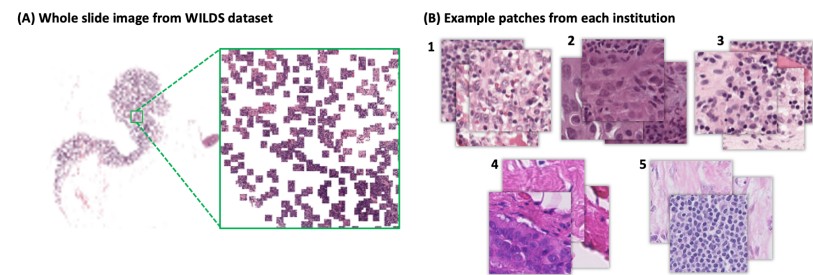

Figure 3: **Representative examples of image patches within the Camelyon17-WILDS dataset illustrating distribution shift.** A. Representative example of patch distribution across a WSI, showing that patches are sampled randomly throughout the tissue sample. B. Representative sample patches from each institution used in the training (1, 2, 3), validation (4) and test (5) sets, showing visual diversity in the images by institution.

## Appendix B. Analysis of convergence

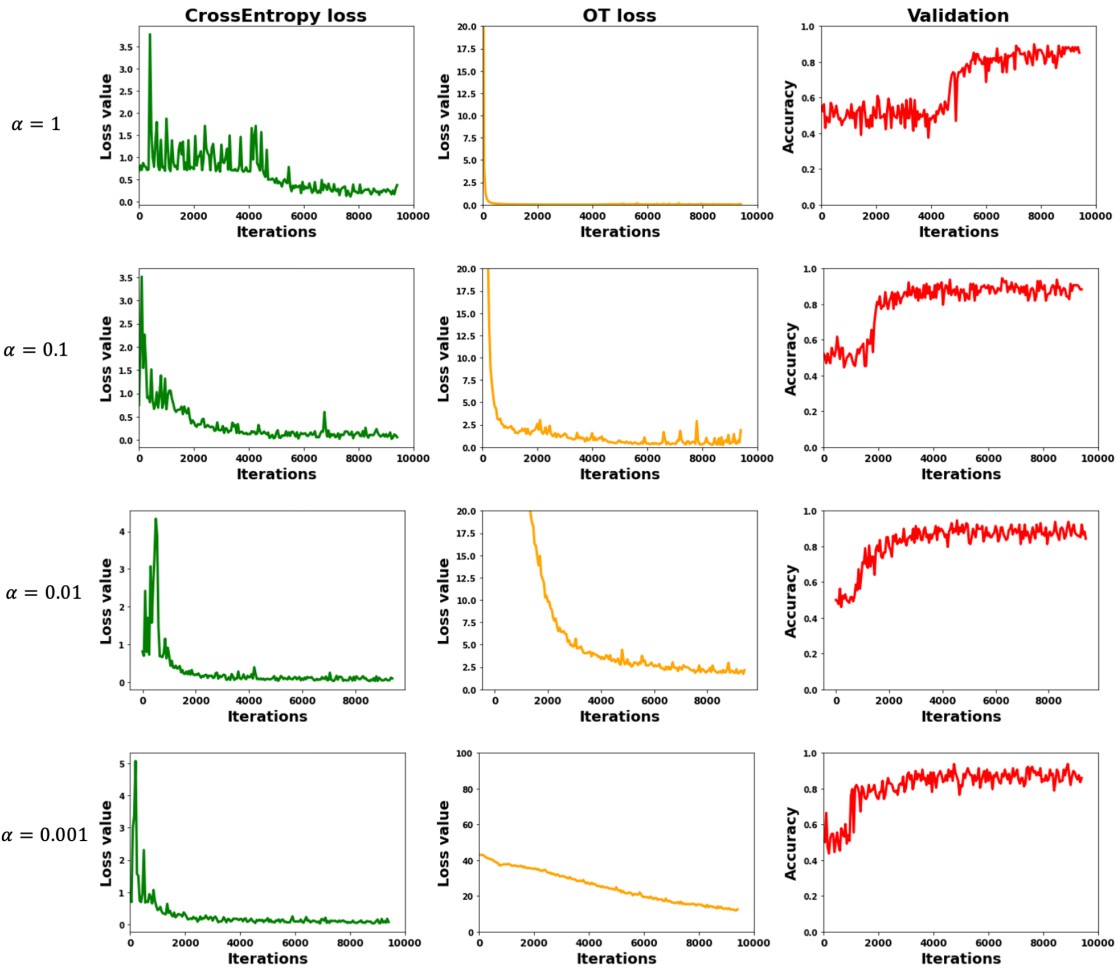

Figure 4: From left to right, plots show CE loss, OT loss, and model performance of the validation set respectively during training. Each row represents different values of $\alpha$.

To optimize the regularization tuning parameter $\alpha$, where $\alpha$ represents the regularized contribution of the OT loss to the overall loss function, we varied $\alpha$ and reported the model performance on the test set, aggregated over 4 experiments initialized with different random number seeds. Our results are summarized in Table 1. When $\alpha$ is very small, the OT loss contribution is minimal and the model does not generalize well, leading to worse model performance. However, at very large $\alpha$, the model struggles to converge, leading to lower classifier accuracy and increased standard deviation of the model performance.

## Appendix C. ROC and AUC values

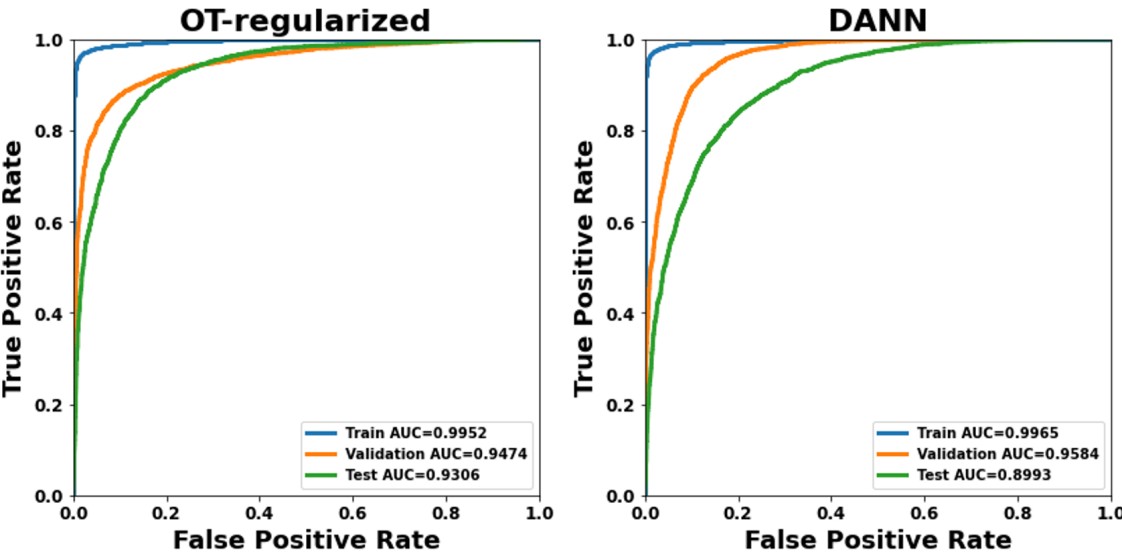

Figure 5: Comparison of Receiver Operating Characteristic (ROC) curves and Area Under the Curve (AUC) values for OT-regularized and DANN method.

The ROC curves for DANN and OT method are plotted with false positive rate (FPR) on the x-axis and true positive rate (TPR) on the y-axis. The AUC values are also reported for both methods, with higher values indicating better performance.

## Appendix D. Aligning target and source distributions after training

One of the approaches that we have investigated was to apply OT to the feature map after training. The idea was to train a model with only cross-entropy loss on source data. Then, forwarding the target data to the feature extractor, aligning the target feature representation with source, and classifying them. To solve this OT problem, we have used Sinkhorn algorithm with the regularization value of 2. As can be seen in Figure 6, the t-SNE projection of features s a good alignment of target data after OT. However, this does not guarantee that the classification will be good, as the validation and test accuracy only slightly improve, with values of 0.855 and 0.711, respectively. This is only slightly better than the ERM approach. Therefore, having a cross-entropy and OT loss function at training time will enhance discrimination power, and therefore prediction accuracy.

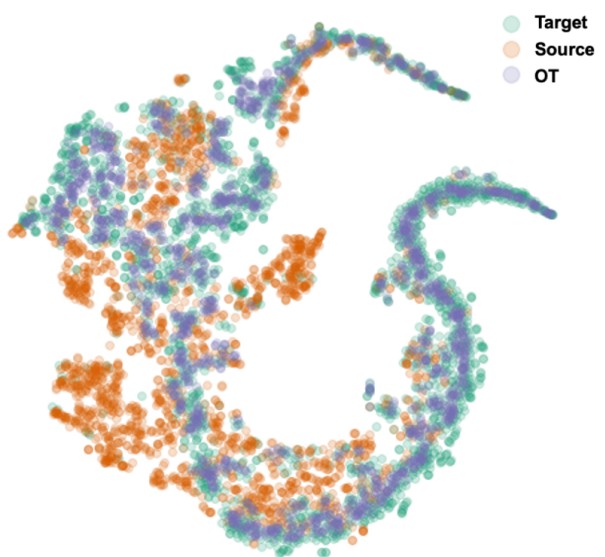

Figure 6: The t-SNE projection of feature embeddings from a random sample of 1000 image patches of target, source, and OT-adapted domains.

## Appendix E. Analysis of feature embeddings

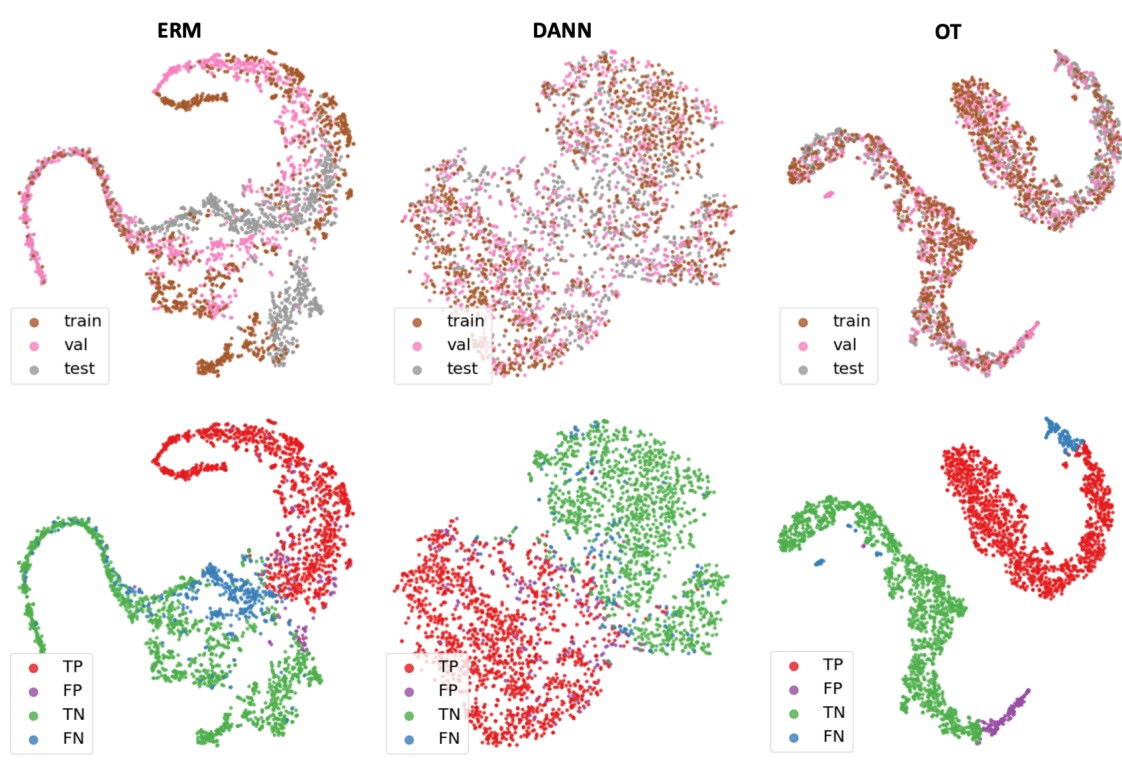

Figure 7: **The t-SNE projection of feature embeddings for EMR, DANN, and OT models.** We demonstrate OT's feature map projection for $(\alpha = 0.1)$. First row highlights each train, validation, and test dataset. Second row represents true positive (TP), false positive (FP), true negative (TN), and false negative (FN).

In Figure 7, we visualize the alignment of feature embeddings between training, validation, and test domains using the baseline ERM method, DANN, and our OT-regularized method. Visually, we observe better alignment between the training, validation and test domains using both DANN and OT. However, OT better separates normal and tumor patches. We also observe visually that using OT loss, incorrect predictions (FP and FN) cluster away from the correct predictions, clearly indicating gaps in the training data diversity.

## Appendix F. permuting institution of validation and test

Table 4: **Model performance compared to DANN method when we use institution 5 as the validation set and institution 4 as the test set (unseen).**
We report standard deviation in the parenthesis

|  | OT-regularized | DANN |
|---|---|---|
| **Accuracy of validation(institution 5)** | 0.877 (0.007) | 0.854 (0.010) |
| **Accuracy of test(institution 4)** | 0.906 (0.006) | 0.861 (0.0124 |

As can be seen in Table 4, the OT-regularized model still performs well when using either institution as the unseen institution. This is an important finding because it suggests that the model is able to generalize well to new institutions even though it was not specifically trained on data from those institutions.

## Appendix G. Stain augmentation

In this section, we analyze the effect of stain augmentation on model generalization. In Table 5, We compare our OT-regularized approach with ERM w/ data aug developed by WILDS and H&E-tailored RandAugment (Faryna et al., 2021). Our findings demonstrate that H&E-tailored RandAugment approach outperform OT-regularized using camelyon17-WILDS dataset. In addition, we integrated OT-regularized approach with H&E-tailored RandAugment and we did not observe any improvement.

Table 5: **Comparison of stain augmentation approaches with OT-regularized.**

| Methods | ERM w/ data aug | H&E-tailored RandAugment | OT-regularized | H&E-tailored RandAugment w/ OT-regularized |
|---|---|---|---|---|
| **Accuracy of validation** | 0.906 (0.012) | 0.914 (0.006) | 0.891 (0.005) | 0.912 (0.005) |
| **Accuracy of test** | 0.820 (0.074) | 0.922 (0.022) | 0.850 (0.019) | 0.924 (0.006) |

