# OpenReview forum: "Domain adaptation using optimal transport for invariant learning using histopathology datasets"
_MIDL.io/2023/Conference — MIDL 2023 Poster_

### Official Review · Reviewer_eK7G · 2023-02-02

**Confidence:** 4
**Preliminary Rating:** 4
**Recommendation:** Oral

**Summary:**

The authors propose a method for domain adaptation to tackle batch effect in computational pathology. The method is based on optimal transport between the distributions of training and validation data, where validation data comes from an unseen center. The testing data comes from an additional unseen center. The authors use the public dataset Camelyon17-WILDS.

**Strengths:**

1. The authors propose a novel method of domain adaptation to tackle batch effect in computational pathology.
2. The authors report results on a public benchmark and compare the performance against multiple baselines from the literature.
3. The authors make the code public.
4. The authors report the performance across 4 reruns.


**Weaknesses:**

1. Multiple times throughout the paper the authors refer to "difference in patients" as a source of domain shift in histopathology, which is flawed on many levels.

“... differences in preparation protocol or scanners can alter the appearance of slides, causing models trained on one institution or patient to fail when generalizing to others.”


"In practice, this would require re-training a model for each new institution and potentially each new patient, which is computationally expensive and time-consuming."


"We believe that these failures may have practical implications in histopathology, as differences in institutions or patients often appear locally in only a fraction..."

In computational pathology, the batch effect/domain shift generally refers to models failing on data because of features that are not diagnostically relevant but differ majorly from the data model was trained on; i.e. a difference in colors between stain providers or a difference in post-processing filters between various scanner vendors (as described in [1] or in papers on domain shift in histopathology cited by the authors in the paper Introduction section). "Differences in patients" in histopathology usually carry diagnostically relevant features, at least in the context of tumor metastasis detection in the lymph nodes dataset(Camelyon17).


2. The authors claim that the proposed method performs better than one of the baselines (DANN) on what the authors refer to as "clumped cancer nuclei phenotype".  It would be highly desirable for the authors to consult with a pathologist to provide a description of those patches.

3. It is not specified which particular centers were in training/validation/testing splits.
4. No analysis of the performance if the centers in training and validation were flipped.
5. The method does not outperform some baselines, however, I believe that it still can be a useful contribution to the community.


[1]M. Aubreville, Mitosis domain generalization in histopathology images — The MIDOG challenge


**Deanonymize Review:**

no

**Detailed Comments:**

1. Do authors consider as an epoch one full run through the training dataset? Was 5 epochs enough for all models to converge?
2. ResNet50 is a relatively large model for this dataset and task. Can authors justify the selection of this particular model? Is it possible that overfitting might have influenced the performance of some methods?




**Paper Type:**

both

**Questions To Address In The Rebuttal:**

1. Correct the statements about "differences in patients" causing batch effect or domain shift.
2. Provide a pathologist description or references to what authors refer to as a "clumped cancer nuclei phenotype".
3. It is not clear which particular centers were used in training, validation, and testing splits. The slides of some of the centers in Camelyon17  were scanned on the same scanner while others were not. This information would be useful for understanding the performance on the splits.
4. Analyze the performance if subsets in training and validation are permuted.

---

### Official Review · Reviewer_cpCd · 2023-02-04

**Confidence:** 3
**Preliminary Rating:** 4

**Summary:**

This paper proposes a new optimal transport loss to penalise differences in domain during training, which leads to more robust performance when evaluating on unseen data from new institutions. During training, there are labeled data from a source domain and unlabelled data from a target domain. The OT loss is used to minimise the differences between features (extracted by classifier head layers) of source domain and target domain.

**Strengths:**

- The ability of adapting classifiers to unseen domains is important for reliable medical AI.
- The paper was written clearly.
- They provides qualitative evaluation to visualise the learnt features with the proposed framework and DANN.
- The code of this work is made publicly available.


**Weaknesses:**

- It seems that the proposed framework trains a classifier with the the source (train) domain and target (validation) domain at the same time during training.  I think it reveals a limitation of the proposed method: it needs to retrain a classifier from scratch for every new domain.
- Although in this paper, the author trained a classifier on domain A and B, and evaluated it on domain C. The results showed that it seems it generalise well. It is still not guaranteed but under the assumption that learning from A and B resulting in invariant features. I think more evaluations or theoretical analysis is required to support this assumption.

**Deanonymize Review:**

no

**Detailed Comments:**

- Better to make the released code also anonymous.
- Some format error in 2.4 after the GitHub link.

**Paper Type:**

validation/application paper

**Questions To Address In The Rebuttal:**

- Why training classifier from scratch with source and target domains, instead of fine-tuning a trained classifier to a new domain?
- Continuing above question, even with the current framework, do you think the performance will improve if fine-tuning on a new domain with OT loss? The result could perhaps validate the hypothesis of learning from two different domains with OT loss can lead to invariant features.

---

### Official Review · Reviewer_s8p6 · 2023-02-04

**Confidence:** 4
**Preliminary Rating:** 3

**Summary:**

The manuscript describes an approach to removing batch effects in histology. The batch effects, often due to staining and imaging techniques adopted at different institutions, are a key challenge in computational analysis in digital pathology. The proposed approach leverages optimal transport to force consistencies in learned representation space between training and testing. The authours tested on classifying normal vs tumour tissue tiles from lymph node metastasis patients. The dataset is wildly tested in domain adaptation benchmarks in ML literature.


**Strengths:**

- The paper has targeted a key bottleneck in the computational analysis of histology images. Solving the batch effect issue could benefit a wide range of tasks in clinical or discovery applications.
- The authours have some positive results on the dataset

**Weaknesses:**

- Lack of comparison with modern baselines for domain adaptation. The benchmarked baselines are relatively old.
- The authors only reported accuracy, not AUC. For a binary classification problem, this is a bit strange.
- Camelyon17-WILDS is the standard test set, but normal vs tumour is a relatively simple and limited task in histology. Also, the 96x96 image tiles are much smaller than the more frequently used 224x224 (or larger) tiles. It is hard to assess the improvement in Camelyon17-WILDS would hold in other applications, especially considering whole-side images.
- The authors claimed their method could work without retraining. It is hard to see why this is the case. The loss function has the cross entropy loss and OT loss, both using a shared network. It isn't clear how retraining can be avoided.
- H&E-tailored RandAugment significantly outperformed the general-purpose OT approaches. Discussion on the added value of the more expensive OT approaches is needed.

**Deanonymize Review:**

no

**Paper Type:**

validation/application paper

**Questions To Address In The Rebuttal:**

- Clarify why the method could work without retraining.
- Providing evidence of generalisability, this could be on other tasks; improved prediction on complex cases (e.g. the author could consider an expanded analysis of figure 6).

---

### Meta-Review · Area_Chair_B7PW · 2023-02-21

**Recommendation:** Accept (Poster)
**Confidence:** 4

**Metareview:**

The paper received overall positive reviews in the initial phase (one borderline, two weak accepts). The rebuttal tried to address most of the concerns. One reviewer raised their score after the rebuttal. Although the work has some weak points which are not evaluated sufficiently (example: population shifts) I believe it provides value to the community and should be accepted with the revision incorporated in the final camera-ready version.